# Quantifying Gait and Posture in Geriatric Inpatients Using Inertial Sensors and Posturography: A Cross-Sectional Study

**DOI:** 10.3390/diagnostics15202578

**Published:** 2025-10-13

**Authors:** René Schwesig, Nicole Strutz, Aline Schönenberg, Matti Panian, Karl-Stefan Delank, Kevin G. Laudner, Tino Prell

**Affiliations:** 1Department of Orthopaedic and Trauma Surgery, Martin-Luther-University Halle-Wittenberg, University Medicine Halle, 06120 Halle, Germanymatti.panian@uk-halle.de (M.P.); stefan.delank@uk-halle.de (K.-S.D.); 2Department of Geriatrics, University Hospital Halle, 06120 Halle, Germany; aline.schoenenberg@uk-halle.de (A.S.); tino.prell@uk-halle.de (T.P.); 3Department of Health Sciences, Hybl Sports Medicine and Performance Center, University of Colorado, Colorado Springs, CO 80918, USA; klaudner@uccs.edu; 4Department of Geriatrics, Jena University Hospital, 07747 Jena, Germany

**Keywords:** gait analysis, posturography, older adults, geriatric ward, instrumented assessments

## Abstract

**Background****/Objectives**: Mobility screening is standard practice in hospitalized geriatric patients, but clinical assessments alone may not fully capture functional capacity and related risks. This study aimed to describe the physical performance (gait analysis, postural stability and regulation) and clinical–functional status (e.g., Tinetti [TIN], Barthel Index [BI]) in geriatric inpatients, and to explore associations between measures from different domains. **Methods**: Fifty-five geriatric inpatients (mean age: 84.3 ± 5.47 years, range: 71–97; 49% female) underwent spatiotemporal gait analysis (inertial sensor system/RehaGait) and posturography (Interactive Balance System). Clinical assessments included TIN, BI, Montreal Cognitive Assessment (MoCA), Geriatric Depression Scale (GDS), Clinical Frailty Scale (CFS), and Numeric Rating Scale (NRS). Gait and postural data were compared with age-, sex-, and height-adjusted reference values. **Results**: Clinical data indicated a low fall risk (TIN: 24), moderate functional independence (BI: 54), and moderate frailty (CFS: 5). Deviations from reference values were more frequent in gait parameters (18/50%) than in postural parameters (6/17%), with postural stability consistently reduced. The largest differences for the geriatric patients compared with the reference gait data were found for stride length, walking speed, double and single support, roll-off angle, and landing angle. TIN showed the strongest correlation with walking speed (r = 0.47, 95% CI: 0.22–0.67), a relationship unaffected by gender (partial r = 0.52). **Conclusions**: Gait assessment revealed greater performance deficits than postural measures in this cohort.

## 1. Introduction

Mobility represents a fundamental determinant of independence, well-being, and quality of life in older adults [1,2,3]. Impairments in gait and posture (stability and regulation) are strongly associated with increased morbidity, loss of independence, institutionalization, and mortality in geriatric populations [4,5]. Among the most serious consequences of mobility limitations are falls, which are highly prevalent in this demographic and often result in injuries, functional decline, and elevated healthcare utilization [6,7,8]. Early identification of gait and postural deficits is thus essential for risk stratification and the development of targeted interventions in geriatric care.

While clinical assessment tools such as the Timed Up and Go test or the Tinetti test (TIN) are widely used to evaluate mobility [9,10], these methods are subject to observer variability and limited in their ability to detect subtle abnormalities in postural regulation and gait performance. As a result, conventional assessments may fail to capture early or complex motor deficits, especially in older adults with multimorbidity or cognitive impairment.

Technological advancements have enabled the development of sensor-based systems and force platform analyses that allow for precise, reproducible, and quantitative measurement of spatiotemporal gait parameters and postural stability and regulation [11]. Instrumented gait analysis provides valuable metrics (e.g., stride length, walking speed, cadence) and gait variability, which are not only sensitive to functional decline but also predictive of adverse health outcomes [12]. Similarly, posturography enables detailed assessment of postural stability and regulation by quantifying center-of-pressure excursions and sway patterns under varying sensory conditions. Objective postural metrics (e.g., medio-lateral sway, synchronization, weigh distribution) provide insights into the integrity of postural subsystems (e.g., visual, vestibular, somatosensory, cerebellar) [13] and are indicative of fall propensity, even when standard clinical tests appear unremarkable. The integration of such quantitative mobility diagnostics into geriatric assessment offers a more nuanced understanding of functional reserve and compensatory mechanisms in aging individuals.

There is an urgent need to incorporate objective, technology-assisted mobility assessments into routine geriatric evaluation. These methods not only enhance diagnostic accuracy but also support individualized care planning and longitudinal monitoring of therapeutic outcomes. As such, this study attempted to address this gap by systematically evaluating gait and postural performance using instrumented assessments in hospitalized geriatric patients undergoing early rehabilitative treatment. The specific aim of the study was to characterize mobility limitations in this vulnerable population and to explore the clinical utility of objective movement analysis as part of comprehensive geriatric care.

## 2. Materials and Methods

### 2.1. Study Design and Patients

This cross-sectional study (Figure 1) was conducted in the Centre for Geriatrics in Southern Saxony-Anhalt (CGC). We included patients over a period of approximately 12–13 months (time interval: 14 May 2024 to 4 June 2025) who received geriatric early rehabilitative complex treatment, a specialized treatment approach for older hospitalized patients with acute illnesses or injuries, recorded under the Operations and Procedures Key (OPS) system 8-550. In the German healthcare system, comprehensive geriatric care within acute geriatric wards (named early rehabilitative geriatric treatment) is coded under the Operation and Procedure Classification System (OPS) 8-550. According to the OPS 8-550 framework, CGC requires the coordinated work of a multiprofessional team, typically comprising geriatric physicians, nursing staff, physiotherapists, occupational and speech therapists, social workers, psychologists, and additional specialists. The shared goal of these professionals is to devise and implement individualized therapeutic programs that strengthen patients’ functional, psychological, and social capacities [14]. Inclusion and exclusion criteria are provided in Table 1. All patients gave written informed consent prior to data collection. The study was approved by the local ethics committee regarding the geriatric assessments (reference number: 2022-026; Date of approval: 24 May 2022) and for the specific movement analysis (reference number: 2025-147; Date of approval: 20 August 2025).

The gait and balance analyses required the patients to stand upright without external assistance and to walk a distance of at least 20 m alone. Additionally, written consent and a minimum age requirement were required in order to participate in the study (Table 1). Exclusion criteria were fundamental orthopedic (e.g., condition after spinal surgery) or neurological (e.g., delirium) limitations that prevented independent test performance (Table 1).

### 2.2. Methods

*Posturography**:* Postural stability, postural regulation, weight distribution and foot coordination were measured using the Interactive Balance System (IBS; neurodata GmbH, Vienna, Austria). The measurement platform of the IBS consists of four independent force plates with a sampling rate of 32 Hz. The captured force-time-signal is converted into a spectrogram using Fast Fourier Transformation (FFT) [13]. Based on the sway intensities across frequency ranges from 0.03–5 Hz, the spectrogram allows assessment of postural regulation, especially the different postural subsystems:F1 (0.03–0.1 Hz): visual and nigrostriatal system;F2–4 (0.1–0.5 Hz): peripheral–vestibular system;F5–6 (0.5–1.0 Hz): somatosensory system;F7–8 (above 1.0 Hz): cerebellar system.

The mapping of frequency bands to specific subsystems (visual/vestibular/somatosensory/cerebellar) should be framed as a functional proxy, not as direct neurophysiological evidence. The assignment of frequency ranges to postural subsystems was obtained with the help of numerous case–control studies [15].

Further detailed information regarding signal transformation and processing, a main characteristic of the IBS, is available in previous publications [16]. Additional to the process parameter, the following motor output parameters were calculated:
stability indicator (ST = postural stability);synchronization (synch = foot coordination);weight distribution index (WDI);forefoot–hindfoot ratio (heel);left–right ratio (left).

Patients performed eight single trials of 32 s each while barefoot for a total time of ≈6–8 min under different test conditions with respect to head and neck position (e.g., straight, rotated 45° to the right and left, up, down), visual input (eyes open or closed) and somatosensory input (standing on foam pads). Detailed information concerning all parameters and test positions can be viewed in previously published work [15,16].

With respect to the execution of the eight test positions, the geriatric patients were instructed to stand upright, with their weight evenly distributed on the two force plates while focusing on a fixed target which was positioned relative to their respective height [13]. The intraobserver reliability of this measurement has previously been validated using asymptomatic subjects [16].

A crucial advantage of the IBS is an available reference database which includes 1724 asymptomatic subjects (age range: 6–97 years) [13]. For our investigated geriatric cohort, we extracted all subjects from this database who were above 70 years old (*n* = 188, male: *n* = 29; female: *n* = 159) to improve the data interpretation. As such, we reported the 50th percentile (median) and the interquartile ranges (25th and 75th percentile) for comparison with our geriatric patients.

*Gait analysis:* In order to capture spatiotemporal gait parameters, a mobile inertial sensor-based system (RehaGait^®^ HASOMED GmbH, Magdeburg, Germany) was used. Its intraobserver reliability and validity [17] has been well documented in earlier studies. In line with Donath et al. [17], each sensor (dimensions: 60 × 15 × 35 mm) contained a 3-axis accelerometer (±8 g), a 3-axis gyroscope (±1000°/s) and a 3-axis compass (±1.3 Gs). Each patient was instructed to walk through a 20 m common hospital corridor (without any obstacles) at a self-preferred speed using their own walking shoes. The first walking trial was performed to adjust to the test conditions. The second trial was used for data collection and statistical analysis. Like IBS, we were able to compare the collected data with age matched reference data. For this purpose, we used the published asymptomatic reference data [18] from all subjects over 70 years (*n* = 132; male: *n* = 60, female: *n* = 72) and also reported the 50th percentile (median) and the interquartile ranges (25th and 75th percentile).

*Geriatric assessments*: Physical, cognitive and mental functions of the cohort were evaluated by using a comprehensive geriatric assessment (Table 2). The TIN and the BI were assessed at the beginning and the end of the geriatric rehabilitative complex treatment.

### 2.3. Statistical Analysis

All statistical analyses were performed using SPSS version 28.0 for Windows (IBM, Armonk, NY, USA). Descriptive statistics (mean, standard deviation, 95% confidence interval (95% CI), interdecile range) were reported depending on the scale level of the parameters.

Mean differences between genders were tested using a one-factorial univariate general linear model for interval-scaled data (e.g., gait and posture). For ordinal-scaled data (e.g., TIN, BI), group differences were tested with Mann–Whitney U-Test, with U statistics and *p*-values reported.

Initially, the significance-relevance level was defined as *p* < 0.05 and partial eta-squared (η_p_^2^ > 0.15) [27]. According to the high number of tests (47) the critical level of significance (*p* < 0.05) was adjusted using the Bonferroni correction. Consequently, the differences between means were considered as statistically significant if *p* values were < 0.001 (0.05/47). Therefore, the significance and relevance levels were ultimately defined as *p* < 0.001 or η_p_^2^ > 0.15. Based on this very conservative approach, we intended to avoid an overestimation of mean differences.

The effect size d (the mean difference between scores divided by the pooled SD) was also calculated for all parameters [28].

Associations between normally distributed metric (Pearson’s correlation) or ordinal (Spearman correlation) parameters were calculated and interpreted as negligible (<0.1), weak (0.1–0.4), moderate (0.4–0.7), strong (0.7–0.9), and very strong (>0.9) [29]. A r > 0.7 and r^2^ > 0.5 (explained variance > 50%) were defined as relevant. To control for the influence of gender, a partial correlation analysis was conducted with gender as the control variable.

The sample size calculation was based on the primary outcome walking speed. Numerous studies indicate that a walking speed of 0.8 m/s (≈2.9 km/h) can be regarded as a cut-off for an increased risk of falling [30,31,32]. As such, the walking speed parameter has an exposed and comparatively well-evaluated importance in the context of gait and posture parameters. The data from Zeh et al. [33] were used to estimate the necessary number of cases, because the authors also used RehaGait and IBS, which markedly increases the validity of the calculation. Accordingly, for a two-sided test of a dependent sample based on an alpha error of 5%, a power (1-beta) of 80%, a mean value of the differences of 0.1 m/s (SD of the mean value differences: 0.17 m/s; d = 0.59), a sample size of *n* = 27 was necessary in order to achieve clinically relevant results. Based on a dropout rate of 20% (*n* = 5.4 ≈ 6), 33 geriatric patients were to be recruited initially.

The presentation of the data is stratified by age and gender in order to offer the possibility for a precise comparison with available reference data for posture [13] and gait [18]. With respect to age, we used 85.0 years as a cut point, oriented on the median of the investigated sample (85.2 years). Additionally, the gait parameters were normalized to body height/mean body height separately for each gender and age group. This approach was based on the finding that gait parameters (e.g., walking speed, cadence, stride length) are strongly dependent on leg length and the leg length is highly correlated with the body height [18,34]:

Equation for time-independent parameters (e.g., stride length (SL))
$${SL}_{cor.}=SL\frac{\emptyset Height}{Height}$$

Equation for time-dependent parameters (e.g., walking speed (WS))
$${WS}_{cor.}=WS\sqrt{\frac{\emptyset Height}{Height}}$$

## 3. Results

### 3.1. Description of the Sample Size

Table 3 contains the clinical characteristics of the investigated population separated between men and women. Female patients were smaller, had lower weight, reported more depressive symptoms according to the Geriatric Depression Score (GDS), and better functional status/independence (Barthel Index, BI).

### 3.2. Postural Stability, Regulation and Weight Distribution

Only synchronization had a relevant gender difference. Female geriatric patients (655 ± 75) demonstrated higher foot coordination than their male counterparts (543 ± 119) (d = 1.16; Table 4).

According to the age- and gender- adjusted reference data, the stability indicator displayed the largest deviations in three out of four cases. Furthermore, in the age range 71.0–85.0 years the female geriatric patients showed a higher activity level in the cerebellar subsystem. For the 85.1–96.5 years age range, the activity level of the somatosensory subsystem was higher in male patients compared to female. The weight distribution (anterior–posterior, heel; medio-lateral, left) was well balanced in male and female geriatric subsamples (exception: females, anterior–posterior, heel: 44.9 ± 7.17% compared to 46.5 to 48.0 in all other samples).

For gait and gender, only for foot height demonstrated a relevant difference for the younger age group (71.0–85.0 years; d = 0.80; Table 5). The male geriatric patients (0.15 ± 0.03 m) displayed higher values than the female geriatric patients (0.13 ± 0.02 m). Moderate effects/differences were observed for foot height (much older adults; d = 0.67) and roll-off angle (older adults; d = 0.62). Male geriatric patients consistently showed a higher foot height, whereas the female geriatric patients had larger roll-off angles (Table 5).

According to the age and gender adjusted reference data, there was a much larger number of values outside the interdecile range (18) compared to the posturographic data (6). The significantly larger part was found in the 71.0–85.0-year-old patients (14) compared to the group of 85.1–96.5 year olds (4). Notably, the gait parameters double support (increased), roll-off angle (reduced) and landing angle (reduced) were most affected (Table 5) as a sign of a reduced postural stability (stability indicator, Table 4) and a reduced ankle mobility or restricted rolling process during walking, respectively.

### 3.3. Association Between Test and Parameters of Different Dimensions

No correlations of practical value (r > 0.7) were found for any clinical parameters during the comprehensive geriatric assessment (Table 6). For clinical and gait parameters, the TIN showed the largest association with walking speed (r = 0.472, 95% CI: 0.215; 0.668), which was unaffected by gender (partial correlation: r = 0.517). On a similar level was the association between TIN and BI (r = 0.451; 95% CI: 0.201; 0.646). The influence of gender was low (partial correlation: 0.473). A notably greater gender effect was detected for Clinical Frailty Scale (CFS) and Geriatric Depression Scale (GDS), where the partial correlation (r = 0.474) was higher than the unadjusted correlation (r = 0.285; 95% CI: 0.007; 0.521). Female patients showed a markedly stronger association between these scores (r = 0.637; 95% CI: 0.321; 0.826) compared to male patients (r = 0.174; 95% CI: −0.232; −0.528).

For clinical and postural parameters, no relevant correlations were identified, with the largest, yet still not relevant, between TIN and F2–4 (r = −0.355; 95% CI: −0.595; −0.056). The effect of gender was again low (partial correlation: r = −0.314).

No relevant correlations were found between gait and postural parameters. All values were below r = −0.415 (95% CI: −0.634; −0.135; Table 6), as observed for the association between landing angle and F5–6, with no notable gender effect (partial correlation: r = −0.419).

## 4. Discussion

In this cohort of geriatric inpatients, clinical assessments indicated generally low risk of fall, moderate functional status, and mid-level frailty, alongside low depressive symptom burden, average cognitive function, and minimal pain. When compared with age- and gender-adjusted reference data, postural stability and regulation showed less pronounced performance deficits than gait parameters, which exhibited the largest deviations from normative values. Marked limitations were observed in indicators related to reduced postural stability and ankle mobility, such as stability index, double support, roll-off, and landing angle. Associations between parameters from different domains were scarce; however, notable links emerged between frailty and depressive symptoms in female patients.

In the present cohort, males had a slightly lower functional status and independence in activities of daily living compared to females. Gender differences were also observed in depressive symptoms, with women reporting a higher symptom burden. TIN scores were comparable between sexes, averaging 24 for both females (SD 17–27) and males (SD 18–27). In a comparable study of geriatric inpatients (*n* = 620, mean age 79.3 ± 8.9 years), Corsinovi et al. [35] reported substantially lower performance levels in both balance (7.2 vs. 12) and gait (6.7 vs. 12) compared to our sample. They also demonstrated that age was a significant predictor of falls, with fallers being older than non-fallers (82.1 ± 7.9 years vs. 78.9 ± 8.9 years; RR = 1.05, 95% CI: 1.01–1.09).

Compared to the asymptomatic reference sample, several performance variables in the geriatric cohort fell within normal ranges, while gait parameters in the 71.0–85.0-year-old patient subgroup showed greater deviations than in the 85.1–96.5-year-old patients, particularly for stride length, walking speed, and roll-off angle. These values also differed from published data for similar cohorts. On average, step length was approximately 44 cm shorter in older men and 37 cm shorter in older women than in the reference group, a finding consistent with the association between lower BI scores and reduced gait performance in this population.

Walking speed in older adults from our cohort (males: 0.79 ± 0.27 m/s, females: 0.76 ± 0.21 m/s) was markedly lower than in an asymptomatic, age-matched reference group (males: 1.21 ± 0.23 m/s, females: 1.20 ± 0.25 m/s) [18]. The values among the matched asymptomatic group are the result of optimal whole-body coordination, which is characterized by minimal temporal-spatial step variability, more stable phase coordination between the limbs, and better synchronized arm swing. Several authors [34,35] also reported higher walking speeds in non-fallers than in fallers among hospitalized geriatric patients. Thus, walking speed may be an appropriate indicator for higher risks of falls, health problems or mortality, with walking speeds below 1 m/s or 0.8 m/s often reported among this population [36,37,38,39]. Regarding spatio-temporal parameters, our investigated patients showed a much higher performance level compared with the patients from Bourgarel et al. [40]:Walking speed [m/s]: 0.71–0.79 vs. 0.47–0.51;Cadence [steps/min]: 96–99 vs. 83–87;Stride length [m]: 0.86–0.94 vs. 0.67–0.69.

In summary, comparison with reference data for gait [18] and posture [13], showed that most deviations occurred in gait parameters (*n* = 6), particularly among older adults (71.0–85.0 years). Walking speed, stride length, double and single support, roll-off angle, and landing angle all deviated substantially from reference values, indicating reduced leg strength (stride length) and ankle flexibility (roll-off and landing angle). Reduced postural stability, as indicated by the stability index, corresponded with prolonged double support and reduced single support. The stability indicator (43.6–51.9) was the only postural parameter which was consistently outside the reference range; except for 85.1–96.5-year-old women (upper threshold: 51.3), all values exceeded age- and gender-specific upper limits (34.8–39.2). from asymptomatic subjects. Apart from 85.1–96.5-year-old females (upper threshold: 51.3), the ST values moved above the age and gender specified upper thresholds (34.8–39.2).

Schwesig et al. [11] prospectively studied nursing home residents (*n* = 146, age 62–101 years) using IBS and RehaWatch to develop a fall index. The most predictive parameters were the peripheral–vestibular system (F2–4), stride time, and variability in landing phase. Gait and postural variability were higher in high-risk fallers, who also had slower walking speeds, similar to our findings and our observed correlation between TIN and walking speed. Schwesig et al. [41] identified the weight distribution index (WDI) and visual/nigrostriatal system (F1) as key predictors in nursing home residents, with a combined postural score (F1 + WDI) outperforming common clinical tests such as the TIN and Timed Up and Go. It should be noted that the IBS is only able to predict (but not measure) these postural subsystems by a FFT of sway indirectly. This kind of diagnostic using surrogate parameters does not replace the comprehensive but very demanding and expensive differential diagnosis concerning posture regulation in several medical disciplines (e.g., neurology, orthopaedics, ophthalmology, otorhinolaryngology, ophthalmology). From a technical point of view, some authors pointed out, that the FFT is only a valid tool for stationary signals [42,43,44]. Algorithms based on Fourier transforms should be used with caution due to the COP can show nonstationary properties [42].

These findings highlight the need for further research, particularly using device-based assessments to evaluate gait and posture [45]. Schwenk et al. [45] emphasized the lack of studies linking frailty with gait characteristics and noted that sensor-based systems (e.g., RehaGait) can capture data across diverse environments, enhancing practicality. Zhang et al. [46] also underscored the value of dynamic measurements, particularly gait variability and confirmed the need for walking distances of at least 10 m. Our data [18] support extending this to 20 m in inertial sensor–based gait analysis to exclude acceleration and deceleration phases and ensure more stable measurements. The value of reference data, central to our approach, was emphasized by Dapp et al. [47], who analyzed gait in 642 community-dwelling older adults (mean age 78.5 ± 4.8 years) using GAITRite and validated results against established geriatric assessments (LUCAS Functional Ability Index, Short Physical Performance Battery). In a comparative inpatient cohort (*n* = 83, age: 83.3  ±  5.88 years, female: *n* = 58), Ollenschläger et al. [48] reported functional gains during hospitalization, with final stride length and walking speed closely matching our results. Independent from age and gender, stride length (0.92 ± 0.21 m) and walking speed (0.70 ± 0.24 m/s) were similar to our results (0.90 ± 0.20 m, 0.76 ± 0.23 m/s). Fränzel et al. [49] observed higher walking speeds and BI scores in a younger hospital sample, though gait was measured over only 10 m. Studenski et al. [50], used a pooled analysis of nine cohorts (*n* = 34,485), and demonstrated that walking speed, which was higher than in our cohort, predicts survival, underscoring its prognostic value as a key spatiotemporal gait parameter.

Our data collection process has several limitations. It is important to note that our study primarily included fitter geriatric patients, as individuals who were non-ambulatory or had severe health impairments preventing gait analysis were not included. Consequently, the findings may not be generalizable to the frailest patient groups. Furthermore, not all participants were able to complete the full set of eight positions required for posturography, which may have reduced the comprehensiveness of the assessment. In gait analysis, a few participants wore open footwear; although the sensors were securely attached, a potential influence on the measurements cannot be entirely ruled out.

For future research, it is important to investigate the relationship between the gait and balance data with respect to risk of fall. The evaluation of these potential relationships are the next step in a longitudinal and prospective study we are conducting.

## 5. Conclusions

Within this selected cohort, mobility analysis as part of a comprehensive geriatric assessment effectively reflected specific deficits between 71.0–85.0-year-old patients and 85.1–96.5-year-old patients in a clinical setting. The absence of strong associations between gait, posture, frailty, mobility, depression, and pain underscores the complexity of each domain and the influence of additional, unmeasured confounders. Integrating instrumented gait analysis and posturography into routine geriatric assessment may enhance diagnostic precision and support more targeted interventions in hospitalized older adults.

Gait and posture analyses in older adults can enable early detection of serious health problems (e.g., fall risk including possible consequences). This early identification then allows for interventions that may prevent fractures and loss of independence, thereby reducing healthcare burdens, and improving the quality of life for older individuals, their families, and their communities.

## Figures and Tables

**Figure 1 diagnostics-15-02578-f001:**
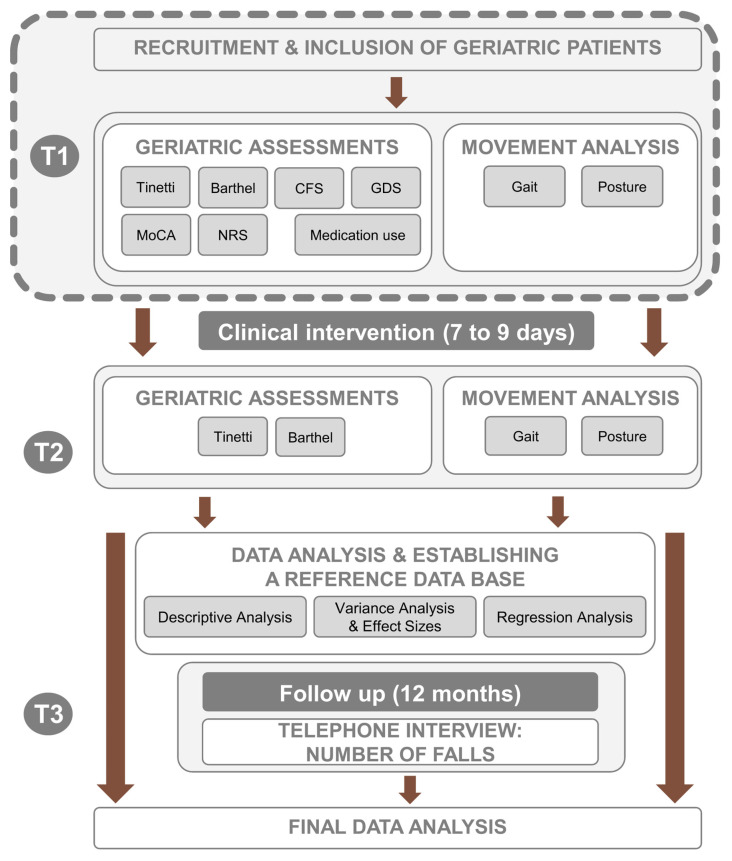
Scheme of study design. T = time point. CFS = Clinical Frailty Scale. GDS = Geriatric Depression Scale. MoCA = Montreal Cognitive Assessment. NRS = Numeric Rating Scale for Pain. Please note that only cross-sectional results of examination one (T1) are part of the manuscript (framed with a dashed line and highlighted in grey).

**Table 1 diagnostics-15-02578-t001:** Patient inclusion and exclusion criteria.

Inclusion Criteria	Exclusion Criteria
Ability to stand upright without external help Ability to walk over 20 m with or without support (walker, walking stick) Written consent for participation in the study Age of ≥70 years	Patients in the end-of-life phase, as assessed by caregivers or by themselves Patients with limb amputations Condition after spinal surgery Delirium

**Table 2 diagnostics-15-02578-t002:** Comprehensive geriatric assessment—description of instrument/assessment and how results were interpreted.

Assessment	Short Description
Tinetti Test (TIN)	The TIN [19,20] is a validated clinical instrument for evaluating balance and gait in older adults. The assessment encompasses measures of static, dynamic, reactive, and anticipatory balance, as well as ambulation and transfer capabilities, providing a comprehensive overview of an individual’s mobility status. The scores range from 1 to 28, where a score of 28 indicates low risk of falling.
Barthel Index (BI)	The BI [21] is an instrument for assessing functional status and the level of independence in activities of daily living (ADLs). It evaluates domains such as eating, bathing, grooming, dressing, bowel and bladder control, toilet use, transfers, mobility on level surfaces, and stair navigation. The scoring system ranges from 0 to 100, with a score of 100 indicating complete independence and a score of 0 reflecting total dependence on assistance.
Clinical Frailty Scale (CFS)	The Clinical Frailty Scale [22,23] is a 9-point, judgement-based tool that classifies older adults from *1 = very fit* to *9 = terminally ill* based on clinical descriptors and functional status, integrating comorbidity, cognition, and level of independence.
Geriatric Depression Scale (GDS)	The Geriatric Depression Scale [24] is a validated 15-item self-report screening tool for depressive symptoms in older adults, with dichotomous responses scored 0–15, where higher scores indicate more symptoms.
Montreal Cognitive Assessment (MoCA)	The Montreal Cognitive Assessment [25] is a screening tool for cognitive function, scored from 0 to 30, with higher scores indicating better cognitive performance.
Numeric Rating Scale (NRS)	The Numeric Rating Scale [26] is an 11-point scale to assess pain intensity in older adults who are able to self-report. A score of 0 reflects no pain and a score of 10 indicates the most pain imaginable.

**Table 3 diagnostics-15-02578-t003:** Demographic, anthropometric and clinical characteristics based on gender at examination 1. Significant-relevant gender differences (*p* < 0.001, η_p_^2^ > 0.15) marked in bold. Results reported as mean ± SD (95% CI) for metric scaled data and as median (interdecile range) for ordinal scaled data.

**Demographic and Anthropometric Data**						
		**male (** * **n** * ** = 28)**	**female (** * **n** * ** = 26)**	**total (** * **n** * ** = 54)**	* **p** *	**η_p_^2^**
Age (years)		83.8 ± 5.96 (81.7; 85.9)	84.7 ± 4.97 (82.6; 86.8)	84.3 ± 5.47 (82.8; 85.7)	0.545	0.01
Height (m)		1.71 ± 0.08 (1.68; 1.74)	1.58 ± 0.08 (1.55; 1.61)	1.64 ± 0.01 (1.62; 1.67)	**<0.001**	**0.39**
Weight (kg)		75.9 ± 12.7 (71.3; 80.6)	64.3 ± 11.9 (59.6; 69.1)	70.2 ± 13.5 (66.6; 73.9)	**0.001**	**0.19**
BMI (kg/m^2^)		26.0 ± 3.58 (24.3; 27.6)	25.9 ± 4.88 (24.3; 27.6)	25.9 ± 4.23 (24.8; 27.1)	0.963	0.00
**Clinical Data**						
		**male (** * **n** * ** = 28)**	**female (** * **n** * ** = 26)**	**total (** * **n** * ** = 54)**	* **p** *	**U**
TIN	total	24 (18; 27)	24 (17; 27)	24 (18; 27)	0.875	355
	balance	12 (8; 14)	12 (6; 14)	12 (8; 14)	0.694	347
	gait	12 (9; 13)	12 (9; 13)	12 (9; 13)	0.380	315
BI		45 (30; 66)	60 (35; 80)	50 (35; 75)	**0.015**	235
CFS		5 (4; 6)	5 (3.8; 6.0)	5 (4; 6)	0.057	262
GDS		1.5 (0; 4.1)	4 (0; 7.3)	2 (0; 6)	**0.007**	212
MoCA		19 (13; 27)	18 (12; 26)	19 (12; 26)	0.815	351
NRS rest		0 (0; 1.1)	0 (0; 1.2)	0 (0; 1)	0.533	348
NRS load		1.5 (0; 3)	2 (0; 4.4)	2 (0; 3)	0.486	338

**Table 4 diagnostics-15-02578-t004:** Posturography—Descriptive comparison based on age and gender (mean ± SD; interdecile range) and analysis of variance (dependent variable: gender). Relevant differences (d ≥ 0.8) and values outside the reference range (interdecile range) of asymptomatic subjects are marked in bold.

	**Male** **(** * **n** * ** = 12)**	**Reference (** * **n** * ** = 20)**	**Female (** * **n** * ** = 10)**	**Reference (** * **n** * ** = 95)**	**Effect Size**
					**d**
**Age Range: 71.0–85.0 years (older Adults)**					
Visual & Nigrostriatal	22.3 ± 7.43 (13.1; 35.8)	19.3 ± 4.21 (13.3; 26.8)	23.9 ± 8.55 (13.5; 40.3)	17.3 ± 6.30 (9.82; 24.6)	0.20
Peripheral– Vestibular	14.2 ± 3.14 (8.96; 18.1)	11.6 ± 2.41 (8.34; 14.8)	14.6 ± 5.23 (7.74; 22.5)	10.4 ± 3.29 (6.88; 15.3)	0.10
Somatosensory	7.79 ± 2.74 (4.76; 11.6)	5.89 ± 1.69 (3.23; 8.17)	8.17 ± 3.78 (5.08; 16.9)	5.59 ± 2.13 (3.42; 8.58)	0.12
Cerebellar	1.70 ± 0.64 (1.00; 2.68)	1.22 ± 0.60 (0.65; 2.47)	**1.90 ± 1.39** (0.80; 5.02)	1.10 ± 0.50 (0.67; 1.60)	0.20
Stability Indicator	**46.6 ± 16.2** (28.4; 71.8)	27.7 ± 8.10 (16.0; 39.0)	**51.9 ± 33.0** (25.8; 126)	25.7 ± 10.1 (15.6; 39.2)	0.22
Weight Distribution Index	5.99 ± 1.90 (2.87; 8.65)	6.12 ± 2.34 (3.84; 10.2)	6.41 ± 0.97 (5.09; 7.85)	6.39 ± 2.70 (3.44; 9.97)	0.29
Synchronization	543 ± 119 (339; 707)	559 ± 182 (224; 769)	655 ± 75 (532; 749)	549 ± 209 (287; 774)	**1.16**
Heel (%)	48.0 ± 6.37 (41.6; 58.2)	48.0 ± 9.97 (33.6; 62.4)	44.9 ± 7.17 (36.9; 56.7)	46.5 ± 9.78 (32.9; 58.3)	0.46
Left (%)	49.8 ± 6.48 (42.6; 62.3)	50.6 ± 4.63 (43.1; 56.5)	48.7 ± 5.81 (38.2; 55.3)	50.5 ± 5.46 (43.9; 56.9)	0.18
**Age Range: 85.1–96.5 years (very older Adults)**					
	**Male** **(*n* = 10)**	**Reference (*n* = 3 *)**	**Female (*n* = 13)**	**Reference (*n* = 36)**	**d**
Visual & Nigrostriatal	22.0 ± 4.17 (16.7; 29.1)	23.0 ± 12.3 (11.4; 35.9)	21.4 ± 4.73 (13.0; 28.7)	20.9 ± 6.57 (13.4; 31.6)	0.07
Peripheral– Vestibular	15.8 ± 3.26 (11.5; 22.0)	13.8 ± 7.75 (9.01; 22.8)	14.9 ± 3.77 (10.2; 22.0)	11.1 ± 4.62 (6.78; 19.4)	0.26
Somatosensory	**7.70 ± 1.45** (5.77; 9.80)	6.93 ± 0.34 (6.64; 7.30)	7.54 ± 1.98 (3.99; 10.3)	6.62 ± 3.83 (3.46; 12.4)	0.09
Cerebellar	1.54 ± 0.44 (1.09; 2.41)	1.44 ± 0.37 (1.08; 1.81)	1.60 ± 0.43 (0.89; 2.18)	1.44 ± 1.14 (0.70; 2.56)	0.14
Stability Indicator	**43.6 ± 10.1** (32.5; 57.8)	30.8 ± 5.01 (25.1; 34.8)	44.4 ± 12.1 (23.7; 60.8)	28.6 ± 18.6 (14.7; 51.3)	0.07
Weight Distribution Index	5.85 ± 1.64 (3.21; 8.89)	7.04 ± 2.06 (5.13; 9.23)	6.78 ± 1.62 (4.84; 9.47)	8.59 ± 3.68 (4.30; 13.9)	0.56
Synchronization	500 ± 204 (167; 739)	423 ± 139 (348; 583)	578 ± 165 (290; 814)	436 ± 177 (146; 640)	0.42
Heel (%)	**49.4 ± 7.60** (35.5; 59.7)	41.3 ± 5.37 (37.2; 47.4)	50.3 ± 9.52 (39.5; 68.4)	39.0 ± 10.5 (24.5; 52.6)	0.11
Left (%)	47.9 ± 5.29 (39.2; 34.5)	51.2 ± 6.85 (43.3; 55.3)	49.1 ± 6.11 (40.4; 58.0)	50.4 ± 6.50 (42.4; 60.0)	0.21

* Due to the small sample size (*n* = 3) median and range were reported. heel: Percentage of weight distribution forefoot vs. hindfoot with description of heel loading. left: Percentage of weight distribution left vs. right with description of left-side loading.

**Table 5 diagnostics-15-02578-t005:** Gait analysis—Comparison based on age and gender (mean ± SD; interdecile range) and analysis of variance (dependent variable: gender). Relevant differences (d ≥ 0.8) and values outside the reference range (interdecile range) of asymptomatic subjects are marked in bold. For bilateral parameters, the left side was used.

	**Male** **(** * **n** * ** = 15)**	**Reference (** * **n** * ** = 47)**	**Female (** * **n** * ** = 10)**	**Reference (** * **n** * ** = 53)**	**Effect Size**
					**d**
**Age Range: 71.0–85.0 years (older Adults)**					
Mean Heights (m)	1.75	1.72	1.58	1.59	
Stride length (m)	**0.93 ± 0.25** (0.54; 1.20)	1.37 ± 0.19 (1.12; 1.61)	**0.91 ± 0.15** (0.72; 1.16)	1.28 ± 0.21 (1.00; 1.53)	0.10
Walking speed (m/s)	**0.79 ± 0.27** (0.40; 1.14)	1.21 ± 0.23 (0.90; 1.51)	**0.76 ± 0.21** (0.40; 1.10)	1.20 ± 0.25 (0.78; 1.50)	0.13
Cadence (steps/min)	99 ± 12.4 (77; 112)	111 ± 9.78 (99; 124)	**98 ± 16.1** (63; 119)	119 ± 10.0 (105; 130)	0.07
Stance phase (%)	65.1 ± 3.61 (59.2; 70.4)	65.5 ± 2.51 (57.3; 63.1)	63.5 ± 5.43 (54.7; 72.5)	59.9 ± 2.95 (56.6; 54.6)	0.35
Double support (%)	**14.5 ± 4.41** (8.80; 21.0)	10.9 ± 2.12 (8.25; 13.4)	**14.3 ± 6.43** (6.22; 27.1)	10.5 ± 2.88 (7.55; 13.9)	0.04
Single support (%)	**35.2 ± 4.24** (29.7; 41.7)	39.2 ± 2.16 (36.4; 42.2)	**35.6 ± 5.83** (25.7; 44.7)	39.5 ± 2.58 (36.3; 43.3)	0.08
Foot height (cm)	**0.15 ± 0.03** (0.10; 0.18)	0.19 ± 0.02 (0.16; 0.22)	0.13 ± 0.02 (0.10; 0.16)	0.17 ± 0.03 (0.13; 0.21)	**0.80**
Roll-off angle (°)	**−33.6 ± 8.90** (−46.8; −19.6)	−65.7 ± 8.68 (−75.9; −53.5)	**−39.8 ± 11.0** (−63.0; −26.7)	−63.5 ± 10.3 (−76.6; −49.5)	0.62
Landing angle (°)	**10.4 ± 6.35** (1.36; 21.2)	27.3 ± 5.77 (20.9; 33.4)	**10.6 ± 4.62** (1.73; 16.3)	22.7 ± 5.05 (15.1; 28.7)	0.04
**Age Range: 85.1–96.5 years (very older Adults)**					
	**Male** **(** * **n** * ** = 10)**	**Reference (** * **n** * ** = 6 *)**	**Female (** * **n** * ** = 15)**	**Reference (** * **n** * ** = 14)**	**d**
Mean heights (m)	1.67	1.71	1.58	1.54	
Stride length (m)	0.94 ± 0.17 (0.74; 1.30)	1.09 ± 0.23 (0.75; 1.40)	0.86 ± 0.22 (0.52; 1.17)	1.02 ± 0.25 (0.73; 1.43)	0.41
Walking speed (m/s)	0.78 ± 0.18 (0.59; 1.17)	0.90 ± 0.24 (0.62; 1.21)	0.71 ± 0.26 (0.30; 1.11)	0.92 ± 0.30 (0.55; 1.39)	0.32
Cadence (steps/min)	98 ± 6.26 (89; 109)	103 ± 10.6 (83; 112)	96 ± 16.3 (68; 119)	111 ± 10.3 (92; 123)	0.18
Stance phase (%)	65.3 ± 3.49 (60.7; 71.3)	63.8 ± 4.67 (58.9; 71.9)	65.4 ± 5.33 (59.3; 74.3)	63.8 ± 5.11 (57.8; 71.8)	0.02
Double support (%)	**16.1 ± 3.96** (10.2; 22.3)	13.6 ± 4.59 (9.93; 21.4)	14.6 ± 5.26 (9.18; 23.6)	13.5 ± 4.17 (8.26; 20.3)	0.33
Single support (%)	34.2 ± 4.33 (27.6; 39.8)	37.7 ± 3.55 (32.7; 41.6)	35.2 ± 5.33 (26.8; 41.3)	37.1 ± 2.69 (32.2; 40.4)	0.21
Foot height (cm)	**0.14 ± 0.03** (0.10; 0.20)	0.18 ± 0.03 (0.15; 0.21)	0.12 ± 0.03 (0.07; 0.17)	0.14 ± 0.02 (0.12; 0.18)	0.67
Roll-off angle (°)	−33.5 ± 7.69 (−45.1; −20.6)	−53.1 ± 8.55 (−62.5; −39.1)	**−37.8 ± 12.0** (−54.1; −18.5)	−56.3 ± 11.6 (−74.5; −38.7)	0.44
Landing angle (°)	**8.22 ± 6.18** (1.94; 21.0)	19.2 ± 5.49 (12.9; 27.9)	10.6 ± 4.97 (3.68; 19.4)	17.0 ± 7.11 (6.05; 28.7)	0.43

* Due to the small sample size (*n* = 6) minimum and maximum were reported in the bracket.

**Table 6 diagnostics-15-02578-t006:** Associations between tests and parameters of different dimensions sorted in descending order based on total correlation (marked in bold).

Parameters	Sample	r	95% CI	Partial Correlation
TIN vs. walking speed	total	**0.472**	0.215–0.668	0.517
	female	0.381	−0.039–0.687	
	male	0.531	0.169–0.767	
TIN vs. BI	total	**0.451**	0.201–0.646	0.473
	female	0.566	0.217–0.786	
	male	0.528	0.181–0.758	
landing angle vs. F5–6	total	**−0.415**	−0.634–−0.135	−0.419
	female	−0.456	−0.731–0.054	
	male	−0.380	−0.697–0.062	
TIN vs. F2–4	total	**−0.355**	−0.595–−0.056	−0.314
	female	−0.410	−0.715–0.028	
	male	−0.218	−0.595–0.237	
CFS vs. GDS	total	**0.285**	0.007–0.521	0.474
	female	0.637	0.321–0.826	
	male	0.174	−0.232–0.528	

## Data Availability

The original contributions presented in this study are included in the article. Further inquiries can be directed to the corresponding author.

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
