# Peer review of "Quantifying Gait and Posture in Geriatric Inpatients Using Inertial Sensors and Posturography: A Cross-Sectional Study"

_diagnostics, 2025, doi:10.3390/diagnostics15202578_

Round 1

Reviewer 1 Report

Comments and Suggestions for Authors

I am delighted to have the opportunity to review an article focusing on gait and postural assessment in older adults. The United Nations Decade of Healthy Ageing (2021–2030) underscores the global priority to enhance mobility, function, and participation in later life. In this context, research on posture and gait in older adults is essential.

Congratulating the team of authors for this article, I have the following comments:

  • Row 18 - clinical status (e.g., Tinetti [TIN], Barthel Index [BI]).

The two recommended scales assess the clinic-functional status, not merely the clinical status. I therefore consider the term clinic-functional status more accurate than clinical status.

  • Row 90 - recorded under the Operations and Procedures Key (OPS) system 8-550

I consider it necessary to provide a brief explanation of the Operations and Procedures Key (OPS) system 8-550, so that any reader is not required to look up the defining elements of this system.

  • In the 1. Study design and patients
    • the time interval during which the study was conducted should be mentioned (duration is 12 months), as well as the date of obtaining study;
    • a precise clarification of how the patients were evaluated is needed, specifically how the scales were administered in relation to the posturographic examination and gait analysis.
  • I consider that Table 2. Demographic, anthropometric and clinical characteristics based on gender at examination would be more appropriately placed in the Results section, especially since it contains statistical data. Additionally, it would be opportune to introduce a column with values for the entire cohort, not just for female / male.
  • For a clearer understanding of the results reported in 3. Association between Test and Parameters of Different Dimensions, a heatmap of the results would be appropriate.
  • In the Discussion section, Row 324, reference is made to Walking speed in older adults from our cohort (males: 0.79 ± 0.27 m/s, females: 0.76 ± 0.21 324 m/s) was markedly lower than in an asymptomatic, age-matched reference group (males: 1.21 ± 325 0.23 m/s, females: 1.20 ± 0.25 m/s) [24]. It might have been helpful to explain that these values are associated with optimal coordination: temporal and spatial step variability is minimal, interlimb phasing is more stable, and arm swing is better synchronized.
  • A few implications for geriatric clinical practice after the study limitations section would be welcome, especially in the context of technical optimization for gait and postural assessment. For example, evaluating gait and posture in older adults enables early identification of fall risk, allowing interventions that prevent fractures and loss of independence, guides effective interventions, reducing healthcare burden, and improving quality of life for older people, their families, and communities.

Reviewer 2 Report

Comments and Suggestions for Authors

The manuscript entitled “Quantifying Gait and Posture in Geriatric Inpatients Using Inertial 2

Sensors and Posturography: A Cross-Sectional-Study”, can be improved toward for the publication to this journal. The study aimed to describe the clinical status and physical performance (gait analysis, postural stability, and regulation). Here are comments.

  1. Figure 1. It should be listed as scheme 1. Figures should be considered as novel findings.
  2. Describe in more details for inclusion and exclusion criteria, why the author selected the criteria.
  3. Should rewriting Equation for time independent parameters into more standard ways.
  4. Should provide more figures that represent more novel analysis or findings.

Reviewer 3 Report

Comments and Suggestions for Authors

This manuscript presents instrumented gait and posturography in hospitalized older adults and relates these measures to standard clinical scales. The topic is timely, the technological approach is innovative, and the workload is substantial (multi-domain assessments in a challenging inpatient population). However, a sharper focus and a more logical, transparent connection between the objectives, methodology, and results would significantly improve the manuscript.

  • The Methods highlight time-dimension elements (7–9 day clinical intervention; planned 12-month follow-up), but the Results are strictly cross-sectional. If the present submission is intended as a cross-sectional study, streamline the Methods to remove or downplay unreported longitudinal components.
  • The current data do not support statements about “predictive value” or longitudinal monitoring. Either report 12-month outcomes and a basic prognostic analysis, or temper language to hypothesis/future work only, and remove predictive framing from the Abstract and Conclusions.
  • The results are hard to interpret because of multiple subgroupings (sex, age strata, patient vs) reference, short-term intervention) and many parameters. Please refactor to a clear primary thread. Move exploratory/secondary subgroup details to the Supplementary material.
  • The mapping of frequency bands to specific subsystems (visual/vestibular/somatosensory/cerebellar) should be framed as a functional proxy, not direct neurophysiological evidence. Please add balanced citations (both supporting and critical) and a brief caveat about overlap and systems integration.
  • Report effect sizes alongside clinical thresholds (e.g., gait speed <0.8 m/s as high risk). Where differences are “statistically significant” but small, clarify whether they are clinically relevant.
  • The Statistical Analysis section specifies using Mann–Whitney U tests for ordinal/non-normal outcomes (e.g., Tinetti, Barthel). However, the Results/Tables do not report Mann–Whitney outputs (U statistics and p-values) nor summarize these variables with median as expected for nonparametric analyses.

The study is worthwhile and novel for the setting, but it needs a sharper analytical focus, alignment of claims with available data, and clearer reporting of reference datasets, statistics, and clinical meaning. After major revision, the manuscript may be reconsidered for publication.

Round 2

Reviewer 3 Report

Comments and Suggestions for Authors

The revision is much improved in clarity and tone. That said, the narrative still places considerable emphasis on the forthcoming longitudinal study and prediction-related material. Because the present work is strictly cross-sectional, the manuscript would read more coherently if it focused on cross-sectional differences and associations, with prediction/longitudinal content moved to a concise ‘Limitations and Outlook’ section. Please minimize references to the ‘next paper’ in the Abstract, Introduction, Methods (including figure legends), Results, and Conclusion; a brief note in the Discussion is sufficient to indicate that prospective analyses will be reported separately. In the Discussion, please also restate the Methods caveat that the band–subsystem mapping is a functional proxy rather than direct neurophysiological evidence, and include one supporting and one cautionary citation for balance.
